# Steel Catenary Riser Fatigue Assessment: Fracture Mechanics Approach Versus *S*–*N* Curve Method

**DOI:** 10.3390/ma17225677

**Published:** 2024-11-20

**Authors:** Niantao Zhang, Caiyan Deng, Wenqiang Zhang, Siyuan Li, Baoming Gong

**Affiliations:** Department of Materials Science and Engineering, Tianjin University, Tianjin 300072, China; 1016208055@tju.edu.cn (N.Z.); dengcy@tju.edu.cn (C.D.); 1020208066@tju.edu.cn (S.L.); gong_bm@tju.edu.cn (B.G.)

**Keywords:** steel catenary risers (SCRs), fatigue assessment, *S*–*N* curve, finite element analysis (FEA), failure assessment diagram (FAD)

## Abstract

In this paper, the fatigue resistance of a full-scale Steel Catenary Riser (SCR) girth weld is investigated using the Strength–Number of cycles (*S*–*N*) curve method based on weld formation quality and fracture mechanics approaches. The test results, presented in the form of *S*–*N* curves, are superior to the design curve E in BS 7608. Compared with the *S*–*N* curve determined by a resonant bending rig, the analytical fracture mechanics, i.e., engineering critical assessment (ECA) based on BS 7910, can provide a rational estimation of full-scale girth welds. For the numerical methods, the short crack growth phase is crucial to improving the accuracy and reliability of the assessment. For the girth weld with a concave root, the geometries of the weld cap are the predominant factors for fatigue life. Although the crack initiation site is always located at the outer surface regardless of the flushed or welded caps, the weld grinding treatment is still effective in promoting fatigue life.

## 1. Introduction

Steel Catenary Risers (SCRs) are free-hanging riser suspended from floating facilities to the seabed in the form of a catenary without buoys devices, as shown in Figure 1 [1]. They have been widely used in offshore oil and gas development in deep-water fields compared to other common riser forms such as flexible pipes, riser towers, and top tensioned risers, due to their reliable performance, easy manufacturing, and convenient installation [2,3,4]. SCRs usually consist of numerous individual segments, i.e., standard-size pipes, which are conveniently and effectively joined by girth welds between adjacent segments. However, the girth weld is usually still the ‘weakest’ link for fatigue resistance due to the inevitable stress concentration, weld defects, and residual stresses. As a result of the continuously dynamic environmental and operational loads, SCRs inevitably suffer severe cyclic vibration along the pipeline axis direction over time during service, making them vulnerable to fatigue failure, even fracture at the point where the ‘weakest’ link is located. Once a part of an SCR is cracked or broken at the location of the girth weld, the oil or natural gas it transports will leak, resulting in serious environmental pollution, ecological disasters, and huge economic losses [5,6].

It is essential to assess the fitness-for-service of SCR structures to ascertain if a known flaw and geometrical discontinuity in the girth weld are likely to fail the structure under cyclic load before the designed in-service life [7], which is normally designed for a life ranging from 10 years to 40 years [8]. There are two widely accepted approaches to fatigue assessment and design of welded joints and components: the *S–N* curve method and the crack propagation method [9,10]. The most common analysis approach for fatigue life prediction is based on the *S–N* curve method, where various fatigue driving parameters may be used, including nominal stress, structural hot-spot stress, and effective notch stress (strain) [11]. The *S–N*-based approach is determined by fatigue testing of the considered welded details and the linear damage hypothesis. The targeted fatigue life at each stress level is determined using the equations recommended in IIW-2259-15:2016 [9], BS 7608:2014 [12], or DNV-RP-C203:2011 [10] with the adjustment at a 97.7% confidence interval. The fatigue design rules of the *S–N* curve are mainly based on data generated from tests on either beams or small-scale plate specimens incorporating the designated weld details, where the verification is performed so that the predicted fatigue life is superior to the target structures with a required safety margin. However, the *S–N* curve method is not able to analyze the fatigue process itself, and the geometrical details and loading histogram are usually omitted or approximated. Therefore, the redundant design based on the *S–N* curve is inevitable. In contrast, the fracture mechanics-based assessment approach assumes the fatigue life of welded joints and components is mainly comprised of the crack propagation from the pre-existed welding defects as described by Paris law and its modifications, whilst the crack initiation is negligible. Fracture mechanics concepts are valuable for assessing the impact of pre-existing welding defects, whereas the S–N curve approach does not consider the actual morphology of these defects. Maljaars et al. evaluated the failure probability of structures under fatigue loading using Linear Elastic Fracture Mechanics (LEFM) [13]. Applying fracture mechanics to the fatigue assessment of offshore pipelines, such as SCRs, is particularly useful for determining whether a sufficient time interval exists between crack detection during in-service inspections and the onset of unstable fracture [14].

In this paper, the full-scale fatigue behaviors of SCR girth welds are tested by a resonant bending rig and plotted in the form of stress versus fatigue life. The welded qualification-related *S–N* curve method according to BS 7608 is compared to the fracture mechanics-based assessment method, i.e., Engineering Critical Assessment (ECA) by BS 7910 and crack growth simulation by FRAC3D incorporated with NASGRO equation, and the accuracy is evaluated in terms of experimental fatigue lives. The comparative investigation of these methods is performed and discussed at the end of this paper.

## 2. Materials and Methods

### 2.1. Material Properties and Welding Process

The pipeline of the SCR in this study is made of API 5L X65 material (Hengyang Valin Steel Tube Co., LTD., Hengyang, China) with an outside diameter of 168.3 mm and a thickness of 18.3 mm. An ER70S-6 material (Jinqiao Welding Materials. Tianjin, China) solid wire and E81T1-Ni1M JH4 material (Jinqiao Welding Materials, Tianjin, China) flux-cored wire with a diameter of 1.2 mm were employed as the Gas Metal Arc Welding (GMAW) root and filler wire, respectively [15]. The chemical compositions of the API 5L X65 and the wires are provided in Table 1 for reference [15]. The material properties for the pipeline steel are as follows: the minimum yield strength, *σ*_s_, is 506 MPa; the tensile strength, *σ*_u_, is 580 MPa; the modulus of elasticity, *E*, is 210 GPa. The fracture toughness of API X65 steel is taken as 109.9 MPa·m^1/2^ according to the software of FRANC3D V8.4.5. During the welding process, the GMAW/Pulse Gas Tungsten Arc Welding (GTAW-P) double-sided root welding process was employed for welding the root welds [15]. Filler metal was utilized in the GTAW root welding and no filler in the GTAW-P root welding process. Subsequently, filling and cover welding were carried out. The detailed parameters of welding processes are provided in Table 1. For the comparative study, weld reinforcements of the SCR girth welds were ground flush by the post-weld treatment in contrast with the as-welded ones at each load level.

### 2.2. Resonance Bending Test for Full Scale Girth Weld Joint

In this paper, a resonant bending rig developed by the Welded Structure Laboratory at Tianjin University was used to study the fatigue life of full-scale girth welds in SCRs, as shown in Figure 2. The pipe, 5530 mm in length, is subjected to excitation with a resonance frequency between 25 Hz and 35 Hz so that each girth weld can undergo alternative loading with stress ratio *R* = 0. Since a standing wave produced by the eccentric mass excites the bending of the pipe, the supporting framework is located at the wave node of the first eigenmode of the pipe, and the weight is carried vertically by air springs. Moreover, the internal water pressure is introduced to achieve higher axial mean stress than zero. When water leakage appears or the pressure drops immediately, a through-thickness crack may exist, and the test is stopped (see Figure 3 for the case).

The tested pipe is joined by two girth welds with a distance of 800 mm plus the width of weldment, as shown in Figure 4. The cyclic stress difference between two girth welds is negligible, as validated by strain gauge measurements. During the testing, eight strain gauges are arranged circumferentially to monitor the axial strain variations positioned 100 mm far from each side of the girth welds, and the illustration of the arrangement is shown in Figure 5. An integrated monitor system is used to acquire the online experimental data, including internal pressure monitored by pressure transducer EPU-2500/YPR-8 (Shanghai Zhendan Sensor Instrument Factory, Shanghai, China) and strain by resistance strain gauge DH5922D (Donghua Test, Jingjiang, China).

The average applied bending stress range was calculated by averaging the strain gauge readings from the adjacent rings and linearly interpolating them to the weld position. Three levels of fatigue stress (high, medium, and low) were used for the tests. Also, the targeted fatigue life at each stress level was determined using the recommendation in BS 7608: 2014, and the adjustment was performed to obtain the failure cycle at a 97.7% confidence interval as described in BS 7608:2014. Post-mortems were performed for both the weld cap and root for crack examination after each test. Moreover, both the measurement of misalignment and macro fractography analysis for the failed strings were conducted to identify the crack initiation and propagation characteristics.

## 3. Fatigue Assessment Based on Fracture Mechanics

The fracture mechanics method describes the entire crack growth history from a small initial crack toward the final failure, which is an indispensable tool once an unavoidable crack or defect is detected and sized. Recently, several standards related to the fatigue assessment based on the fracture mechanics principle have been widely used in ocean engineering, including BS 7910:2013 [16], DNV C203:2011 [10], and IIW-2259-15:2016. In this section, the study of fatigue resistance assessment of the SCR in terms of a numerical model by NASGRO and the analytical failure assessment diagram method is performed.

### 3.1. The NASGRO Model by FRANC3D

To achieve a reliable prediction for the growth of a fatigue crack in the high cycle regime, crack propagation behaviors with both scales comparable to the microstructural grain size and the local plastic zone size are essential. The well-known NASGRO equation for fatigue life prediction based on LEFM is an analytical crack growth rate equation that incorporates several features observed in real materials, such as sensitivity to near-threshold and near-critical growth, sensitivity to the stress ratio, and small crack sensitivity [17,18,19]. The basic formula of the NASGRO crack growth rate equation is expressed as follows:(1)$$\frac{da}{dN}=C{\left[\left(\frac{1-f}{1-R}\right)\Delta K\right]}^{n}\frac{{\left(1-\frac{\Delta K_{\mathrm{th}}}{\Delta K}\right)}^{p}}{{\left(1-\frac{\Delta K_{\max}}{K_{c}}\right)}^{q}}$$
where C, n, p, and q all are empirical constants; f is the crack opening function associated with plasticity-induced crack closure, which has been defined by Newman [20]; $\Delta K_{\mathrm{th}}$ is the threshold stress intensity factor range below which there is no crack growth. It is approximated by the following empirical formulae:(2)$$\Delta K_{\mathrm{th}}=\left\{\begin{array}{c} \Delta K_{1}^{*}{\left[\frac{1-R}{1-f}\right]}^{\left(1+RC_{\mathrm{th}}^{p}\right)}/{\left(1-A_{0}\right)}^{\left(1-R\right)C_{\mathrm{th}}^{p}}\;\;R\geq 0\\ \Delta K_{1}^{*}{\left[\frac{1-R}{1-f}\right]}^{\left(1+RC_{\mathrm{th}}^{m}\right)}/{\left(1-A_{0}\right)}^{\left(C_{\mathrm{th}}^{p}-RC_{\mathrm{th}}^{m}\right)}\;\;R<0 \end{array}\right.$$
and $\Delta K_{1}^{*}$ can be calculated as follows:(3)$$\Delta K_{1}^{*}=\Delta K_{1}{\left[\frac{a}{a+a_{0}}\right]}^{1/2}$$
where $C_{\mathrm{th}}$ is an empirical constant, $\Delta K_{1}$ is the observed threshold for a high-stress ratio, and $a_{0}$ is a small crack parameter, also known as the intrinsic crack length (usually 0.0381 mm). $K_{c}$ is the fracture toughness that should be chosen via the plane-strain fracture toughness, $K_{1c}$, the plane-stress toughness, $K_{1e}$, or calculated as follows:(4)$$K_{c}=K_{\mathrm{Ic}}\left(1+B_{k}e^{-{\left(A_{k}t/t_{0}\right)}^{2}}\right)$$
where $t_{0}=2.5{\left(K_{\mathrm{Ic}}/\sigma_{\mathrm{ys}}\right)}^{2}$, $A_{k}$ and $B_{k}$ are empirical constants. The thickness, t should be specified on the basis of engineering judgment. In this work, the plane-strain fracture toughness, $K_{\mathrm{Ic}}$, is chosen. The parameters required in the NASGRO equation are provided in Table 2.

### 3.2. FAD Method in BS 7910

Failure Assessment Diagram (FAD) and Crack Driving Force (CDF) approaches are two basic engineering assessment methods that are aimed at providing conservative assessments of the severity of crack-like flaws. This technique originates from the work of the British Central Electricity Generating Board (CEGB) and its R6 assessment procedure. The FAD approach is recognized as an efficient methodology to assess the potential interaction between brittle fracture and plastic collapse of loaded structural components that contain cracks, where a rough geometry-independent failure line is constructed by normalizing the CDF by the material fracture resistance [21]. More specifically, the vertical axis of the FAD compares the applied loading, in terms of the corresponding CDFs normalized by the fracture toughness of the material; the horizontal axis is the ratio of the applied primary load to that required for plastic collapse. The assessment of the component is then based on the relative location of a geometry-dependent assessment point with respect to this Failure Assessment Line (FAL), which is considered safe as long as the assessment point lies within the area below the FAL, and potentially unsafe if it is located on the line or outside the shaded area, as illustrated in Figure 6 [22]. Many studies have been done to verify the efficiency of the FAD concept in evaluating the mechanical integrity of components with flaws. The BS 7910 is the most widely used engineering code to evaluate the acceptability of flaws in metallic structures [23]. In Figure 6, Option 1, presented as the function of $L_{r}$, is recommended for general use in cases where knowledge regarding the material properties is limited. It is a conservative procedure that is relatively simple to employ and does not require detailed stress–strain data for the materials being analyzed; Option 2 is based on the use of a material-specific stress–strain curve; Option 3 uses numerical analysis to generate a FAD and is not confined to use with materials showing ductile tearing.

(1)Option 1 of failure assessment procedures: basic level
(5)$$f\left(L_{r}\right)=(1-0.14L_{r}^{2})[0.3+0.7\exp(-0.65L_{r}^{6})]\;\;\;L_{r}\leq L_{r,\max}$$(2)Option 2 of failure assessment procedures: standard level
(6)$$f\left(L_{r}\right)={\left(\frac{E\varepsilon_{\mathrm{ref}}}{L_{r}\sigma_{s}}+\frac{L_{r}^{3}\sigma_{s}}{2E\varepsilon_{\mathrm{ref}}}\right)}^{\frac{1}{2}}\;\;\;L_{r}\leq L_{r,\max}$$
where $\varepsilon_{\mathrm{ref}}$ is the reference strain.(3)Option 3 of failure assessment procedures: advanced level
(7)$$f(L_{r})=\sqrt{\frac{J_{\mathrm{el}}}{J_{\mathrm{el}}+J_{\mathrm{pl}}}}\;\;\;L_{r}\leq L_{r,\max}$$
where $J_{\mathrm{el}}$ and $J_{\mathrm{ep}}$ are the elastic and plastic components of *J*-integral, respectively.

The cut-off line of the FAC for the above procedures is defined as:(8)$$L_{r,\max}=\frac{\sigma_{s}+\sigma_{u}}{2\sigma_{s}}$$

When the value of $L_{r}$ exceeds the $L_{r,\max}$, the value of $K_{r}$ is taken as zero. In this paper, Option 1 obtained from the bottom-bound fitting of the FAD curve family generated using Equation (5) for various material constitutive relations was employed.

## 4. Result

### 4.1. S–N Curve and Comparison with BS 7608

The test results obtained from the full-scale pipes are presented in Table 3 and Figure 7, respectively. In this study, the fatigue life was defined as the number of cycles required to produce a through-wall fatigue crack or greater than 10^7^ (referred to as run-out). The design curve D in BS 7608:2014 is designated as the targeted qualification of outer diameter for the tests. Meanwhile, the design curve E in BS 7608:2014 with the adjusted value at a 97.7% confidence interval was also plotted in Figure 7 for comparison. The test results are generally better than the BS 7608: 2014 Class E design *S–N* curve. Both the low (Δσ = 68.9 MPa) and medium (Δσ = 103.4 MPa) stresses cannot produce final failure until an endurance larger than 3.2 × 10^7^ cycles and 1.2 × 10^7^ cycles, respectively. Therefore, only the tests at high loading levels are studied further for comparison with other methods. More specifically, it is found in Figure 8 that fatigue crack initiated at the toe of the convex weld root and grew through the wall thickness for the external polished pipe (Specimen H-1), while the cap weld toe was the crack initiation site and then propagated perpendicular to the wall thickness for the as-weld pipe (Specimen H-2). In contrast, it is found in Figure 9 that all the cracks initiate at the external surface for the flushed and as-welded girth due to the concave weld roots, and many more failure cycles are achieved. Therefore, the geometrical concentration introduced by the girth weld root is predominant, and the crack initiation site depends on weld root convexity. Moreover, although the reinforcement flush treatment can extend the fatigue life for both convex and concave weld roots, Specimen H-1 still exhibits an unexpectedly shorter fatigue life compared to specimen H-2. This is because crack initiation at the internal surface of the pipe shields the life extension effect of the cap flush. The crack initiation transitions from the external surface to the internal surface of the pipe, which is more difficult to detect during routine inspection and may result in catastrophic failure of the structures.

### 4.2. Numerical Prediction Based on the Crack Growth Model

In order to simulate the crack growth using linear elastic fracture mechanics (LEFM) and evaluate the fatigue life of the SCRs, Finite Element Analysis (FEA) was performed on the three-dimensional model of the pipeline with the combination of the commercial software ABAQUS 2022 and FRANC3D V8.4.5 [24,25]. FRANC3D, a fracture mechanics tool, is designed to analyze crack growth in Finite Element (FE) models generated by ABAQUS. Its primary function involves inserting and extending crack tip elements, leveraging quadratic order elements, such as 20-node solid quadratic elements, in three-dimensional models to accurately represent the stress fields near the crack tip. ABAQUS serves as the solver for the models modified by FRANC3D, facilitating the calculation to generate the output database corresponding to various load conditions and crack geometrics. Subsequently, the Stress Intensity Factors (SIFs) along the crack front are calculated based on the displacement field extracted from the output database using the M-Integral in FRANC3D. This process also enables the determination of the next crack growth increment, allowing for continuous extension of the crack.

The model is partitioned for ease of meshing and locating of the crack. Detailed focused meshes are required around the crack tip for accurate contour integral evaluation. Abaqus integrates around the ring of element (contour) enclosing the crack tip to determine the Stress Intensity Factors (SIFs). Contour integrals are evaluated for multiple rings of elements surrounding the crack tip node. The first contour is formed from elements directly connected to the crack tip node. Each subsequent contour is created by offsetting one element away from the previous contour. The crack aspect ratio (crack depth/crack half-length, *a*/2*c*) is assumed as 0.1 based on recommendations in [26]. Cracks in SCRs typically start from a long shallow surface flaw situated at the interface of weld and parent metal, and then grow gradually as a semi-elliptical flaw until it penetrates the wall thickness (see Figure 8).

#### 4.2.1. Equivalence of Load Condition

Online experimental data of the stress state was acquired from the integrated monitor system via the strain gauges arranged circumferentially as shown in Figure 5. The average of the maximum stress range recorded by each strain gauge throughout the test is shown as a stress circle in Figure 10. Compared with the numerical magnitude of the stress range, the difference in the stress range at different strain gauge positions is negligible. Therefore, for specimen H-1 and specimen H-2, the load conditions at both ends of the girth weld in the resonance bending test condition are equivalent to the stress ranges of 178.6 MPa and 174.8 MPa, respectively. The equivalent load conditions are later employed in the simulation analysis of crack propagation.

#### 4.2.2. Geometry Simplification of the Pipeline

The geometric model should be determined according to the actual geometry of the pipeline for accurate fatigue life prediction. Since the weld profile in terms of weld reinforcement height, toe radius, and toe angle varies along the location of the grith weld, it is thus impractical to model the actual geometrical discontinuity in the numerical analyses. However, the effects of weld reinforcement cannot be omitted when a shallow surface flaw (z/B<0.2) is located around the stress concentration of the weld toe or weld root. Therefore, a plain pipeline model instead of the actual geometry of the SCR is employed, while a stress correction factor, $M_{k}$, multiplying the actual axial stresses monitored by strain gauges is applied to compensate for the stress concentration effect caused by the geometric discontinuity at both weld cap and weld root toes, which is regarded as geometry simplification as shown in Figure 11. More specifically, when the real structure of the weld profile is neglected in the process of model simplification, the corresponding axial stress P is adjusted by multiplying it with $M_{k}$ calculated via empirical formula. Finally, the new geometric model, which is equivalent to the original, has been constructed, and it provides considerable convenience for both modeling and insertion of initiation in the subsequent subsection. $M_{k}$ is defined according to BS 7910: 2013 as follows:(9)$$M_{k}=v{\left(z/B\right)}^{w}$$
where v and w have different values corresponding to flaws at the toes of full penetration or attachment welds, respectively. $M_{k}$ and the detailed size of the macrographs of the weld are listed in Table 4.

#### 4.2.3. Modeling and Insertion of the Initial Crack

To simulate the crack growth and evaluate the fatigue life using the NASGRO equation, an initial surface crack was introduced into stress concentration zones, such as the toe of the weld root (H-1) or weld cap (H-2), in accordance with the macrograph of the weld cross-section. The numerical analyses were conducted for six cases as tabulated in Table 5 and illustrated in Figure 12. Only half of the FE model was modeled using eight-node linear brick elements (C3D8) due to the symmetry. The process of geometric simplification has been detailed in Figure 11, and the transition mesh was applied to the welded joint section of the global model as shown in Figure 12b. As shown in Figure 12c, a sub-model was separated from the global model, and the corresponding boundary conditions were introduced into FRANC3D, where half of the long shallow crack was inserted with the sizes of d = 1.00 mm, e = 0.11 mm, and $r_{c}$ = 0.10 mm, respectively.

#### 4.2.4. Simulation of Fatigue Crack Growth

There are several different criteria to calculate the direction of crack growth, and one of the most widely used criteria is the maximum tangential stress (MTS) criterion proposed by Erdogan and Sih [27]. The results indicate that the value of $K_{I}$ is larger than $K_{\mathrm{II}}$ and $K_{\mathrm{III}}$ as shown Figure 13, suggesting that mode-I (opening mode) predominantly governs the crack growth process [28]. In FRANC3D, the SIFs evolution, as shown in Figure 13b, was constructed using the increment of crack depth and the calculated SIF values at the crack front nodes. The crack distance is normalized along the crack front as shown in the upper inset of Figure 13a, where point G represents the intersection point of the symmetric boundary and the crack front, i.e., the intermediate point of the complete crack, and point H represents intersection point of the inner surface of the pipeline in specimen H-1 (the outer surface of pipeline in specimen H-2) and the crack front as shown in the lower inset of Figure 13a. The SIFs result shows the crack type is the opening mode under the uniaxial load condition, and the value of $K_{I}$ is above the threshold values $\Delta K_{\mathrm{th}}$. The propagation path and SIFs distribution along the crack front can be seen from Figure 13b. Also, as the crack grows, $K_{I}$ of point G becomes larger. As a result, at the final step of crack propagation through the thickness, the SIFs reach their maximum value.

Figure 14 illustrates the trend in fatigue life and SIFs at point G as crack depth increases. During the crack growth process, when the crack depth exceeds 8 mm, the SIFs of specimen H-1 are distinctly higher than that of specimen H-2, and the gap between them progressively widens. The higher SIFs lead to a reduction in fatigue life, as illustrated in Figure 14, where specimen H-2 shows a longer fatigue life compared to specimen H-1. This finding is consistent with the results of the resonance bending test for the full-scale girth weld joint, where the fatigue life extension effect of the reinforcement flush treatment is masked.

#### 4.2.5. Fatigue Life Prediction

As discussed before, it is found that the compensation for the stress concentration effect by M_k_ has a significant influence on the axial load, as shown in Figure 15. The fatigue life of the pipeline under different conditions is summarized in Table 6. Comparing the calculated total number of cycles to failure against that without the stress correction factor, a noticeable variation is found in the predicted fatigue life for specimens H-1 and H-2, which decrease by approximately 80.4% and 98.4%, respectively, at R = 0.1.

### 4.3. Assessment by FAD

For the structural assessment based on the FAD method, a pipe riser with a longitudinal semi-elliptical crack oriented in the axial direction of the pipe on its outer or internal surface was idealized. The length and depth of the semi-elliptical crack were characterized by 2*c* and *a* with the assumed aspect ratio *a/c* = 0.1, which is consistent with the initial crack in the numerical simulations. Herein, the welding misalignment was also taken into account.

#### 4.3.1. Stress Intensity Factor Solution

The fracture mechanics-based Engineering Critical Assessment (ECA) approach assumes that a flaw can be idealized as a sharp-tipped crack that propagates in accordance with the law relating the crack growth rate, da/dN, and the range of stress intensity factor, ΔK, for the material containing the flaw, where the correlation law follows the Paris formula as follow:(10)$$\frac{da}{dN}=C{\left(\Delta K\right)}^{m}$$
where C and m are a constant that depends on the material and the applied conditions, including the environment and cyclic frequency. Herein, the two-stage fatigue crack growth relationship is employed, with stage one near the crack growth threshold and stage two beyond it. The values of the constant $C_{1}$ and $m_{1}$ in stage one are 2.1 × 10^−17^ and 5.1, respectively, and the values of the constant $C_{2}$ and $m_{2}$ in stage two are 1.29 × 10^−12^ and 2.88, respectively. The SIF solution for fatigue assessments is given by:(11)$$\Delta K_{I}=Y(\Delta \sigma )\sqrt{\pi a}$$
(12)$${\left(Y\Delta \sigma \right)}_{p}=Mf_{w}\{k_{\mathrm{tm}}M_{\mathrm{km}}M_{m}\Delta \sigma_{m}+k_{\mathrm{tb}}M_{\mathrm{kb}}M_{b}[\Delta \sigma_{b}+\left(k_{m}-1\right)\Delta \sigma_{m}$$
where the subscript *p* indicates the primary stress and the subscript *m*, *b* indicates the membrane stress and bending stress, respectively. Only the effect of membrane stress on fatigue life is considered. The influence of misalignment detailed in Table 7 is taken into account by $k_{m}$ in Equation (14) and expressed by:(13)$$k_{m}=\frac{\Delta P_{m}+\Delta \sigma_{s}}{\Delta P_{m}}=1+\frac{\Delta \sigma_{s}}{\Delta P_{m}}$$
(14)$$\frac{\sigma_{s}}{P_{m}}=\frac{6e}{B_{1}\left(1-\nu^{2}\right)}\left[\frac{1}{1+{\left(B_{2}/B_{1}\right)}^{1.5}}\right]\sigma_{s}/P_{m}<1$$

#### 4.3.2. Fatigue–Fracture Assessment

The assessment procedures were implemented using the commercial software CrackWISE 5.0, which allows both the fatigue and fracture assessments to be performed. The assessment in terms of fatigue life versus crack depth is shown in Figure 16, and the corresponding parameters and results are listed in Table 8. It was found that the fatigue life tends to flatten out as the crack grows, and the real-time crack size after each crack depth increment, Δa, is evaluated for fracture instability or plastic collapse, as shown in Figure 17, to ensure its integrity. The structure is considered to have failed once the assessment point $\left(L_{r},K_{r}\right)$ exceeds the area below the FAC, as designed using Option 1 of BS 7910:2013, then the number of load cycles is regarded as the failure life. Meanwhile, the loci of the assessment point under different load conditions can be found in Figure 17, and the final failure size of the crack and fatigue life are summarized in Table 8.

### 4.4. Comparative Study of Assessment Methods

Based on the results of the different assessment methods described above, the percentage deviation from the experiment is obtained and tabulated in Table 9, where a negative sign indicates lower than the experimental results and a positive sign indicates higher than the experimental results. It is found that when the stress concentration effect is neglected, the numerical prediction results are closer to the experimental results, with a better percentage deviation of even +2.38%. Therefore,$M_{k}$ has a significant effect on the fatigue life of specimen H-1 and specimen H-2, leading to a decrease of approximately 79.93% and 98.67%, respectively, at R = 0.1 compared to the experimental results. There is a relatively high deviation for specimens H-3 and H-4, and the percentage deviations with the stress ratio of 0.1 are −52.09% and −81.76%, respectively. When the $M_{k}$ is considered, the deviation is even greater, almost 100 percent. As for the FAD method, the percentage deviation remains at −47.29% and −38.85% for specimen H-1 and specimen H-2, respectively, and is −66.78 for both specimen H-3 and specimen H-4.

## 5. Summary

In this work, the full-scale fatigue behaviors of SCR girth welds are tested by a resonant bending rig. The test results are presented in the form of *S–N* curves and are superior to the design curves E in BS 7608. The cap flush is still an effective post-weld treatment to improve the fatigue behavior of full-scale girth weld, and its benefit is dependent on the original weldment profiles. Moreover, the fatigue assessment methods based on fracture mechanics were performed, and the comparison with the welded qualification-related *S–N* curve was studied. The summary is concluded as follows:(a)The geometrical concentration introduced by full-scale girth weld is predominant for fatigue behaviors, and the competition for crack initiation exists between the toes of the weld root and cap, which depends on the weld root convexity;(b)The analytical results based on fracture mechanics are able to provide a rational estimation of full-scale girth weld, although the conservative prediction is still not trivial. The $M_{k}$, which refers to the stress correction factor recommended, may overestimate the three-dimensional stress intensity factor for internal surface cracks located at the girth pipe root toe. To improve the accuracy, the new formula of $M_{k}$ for both weld cap and root toes shall be determined by three-dimensional finite element simulation;(c)For the numerical assessment (NASGRO equation) methods, the assessed fatigue life is relatively lower than the experimental results due to the neglected short crack growth. However, for the well-manufactured welded joint, it is widely recognized that the crack initiation or short crack growth phase amounts to most of the fatigue life. However, the present fracture mechanics approach is still under development to incorporate the proper short crack propagation theory into the fatigue assessment to improve their accuracy and ratability;(d)For internal surface cracks located at the girth pipe root toe, to assess the impact of various parameters on the $M_{k}$, three-dimensional finite element models are developed. These models take different types of welding misalignment, a range of variable-wall-thickness ratios, and initial crack depths into account. The findings reveal that the stress intensity correction factor Mk tends to increase with the rise in misalignment and variable-wall-thickness ratio.

## Figures and Tables

**Figure 1 materials-17-05677-f001:**
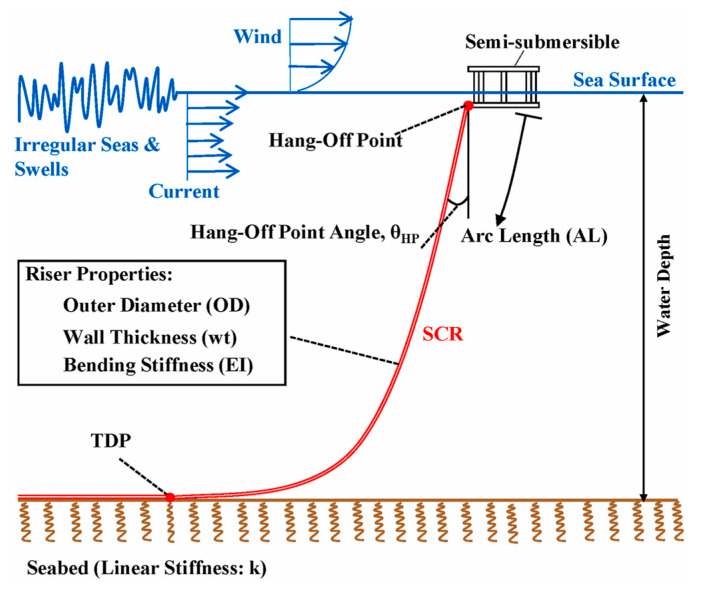
Overall view of the floating system with SCR [1].

**Figure 2 materials-17-05677-f002:**
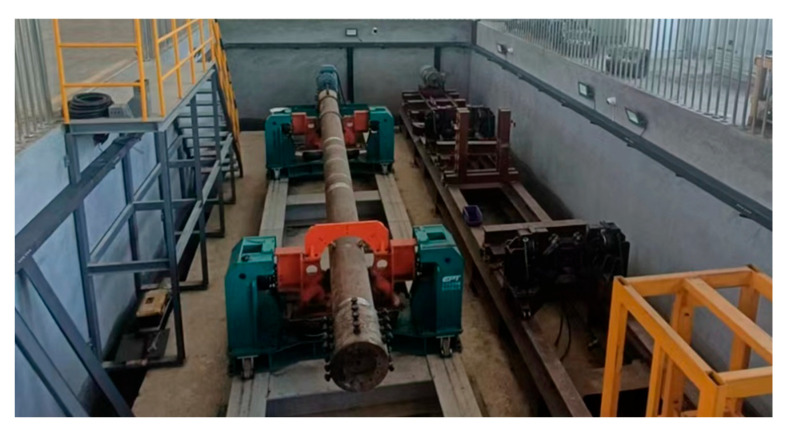
Resonance bending fatigue testing platform, Tianjin University (equipment no.: GWT0614; dimensions: 15 m × 1.3 m × 2.0 m; range of the test pipeline diameter: 168.3–355.6 mm; range of the test pipeline length: 6–12 m; frequency: 0–25 Hz).

**Figure 3 materials-17-05677-f003:**
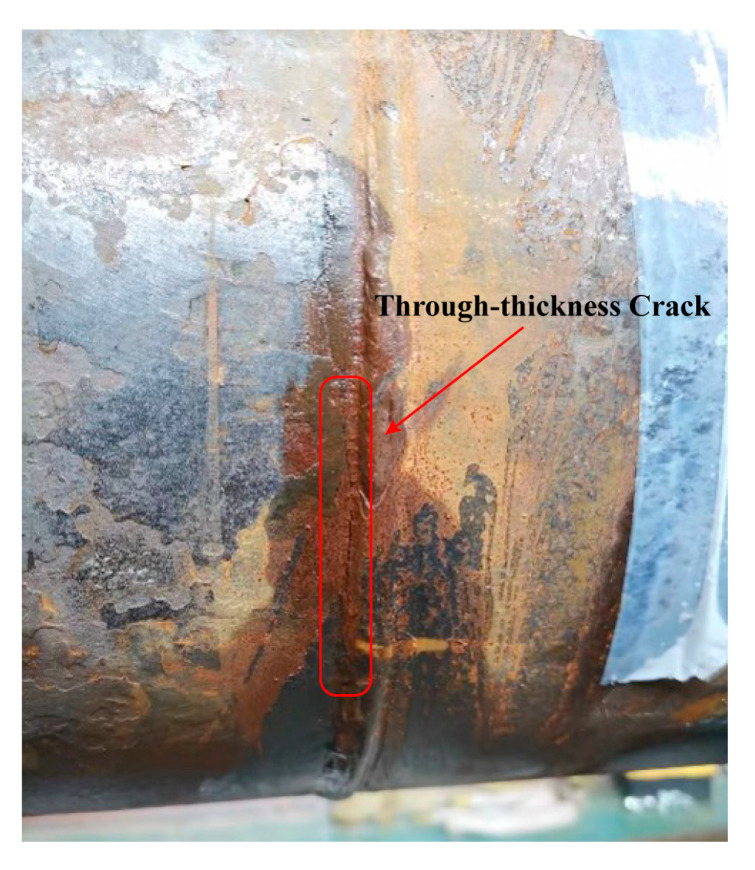
The occurrence of a through-thickness crack after internal water pressure leak.

**Figure 4 materials-17-05677-f004:**
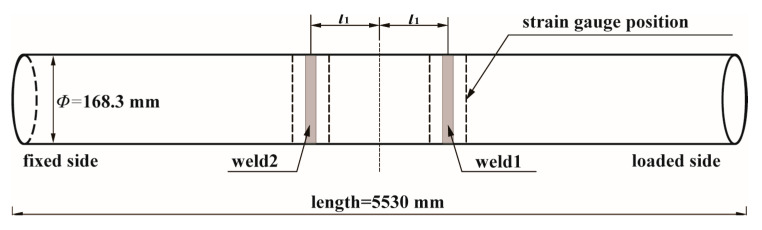
Schematic diagram of resonance bending fatigue test of full-size SCR (*l*_1_ = 400 mm).

**Figure 5 materials-17-05677-f005:**
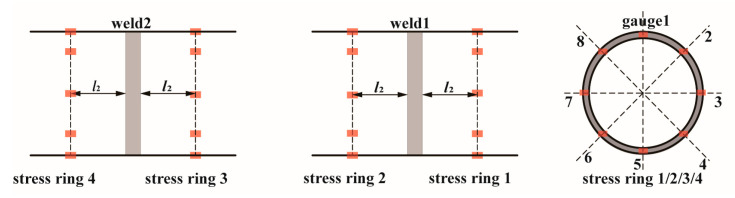
Schematic diagram of strain gauge arrangement scheme (*l*_2_ = 100 mm).

**Figure 6 materials-17-05677-f006:**
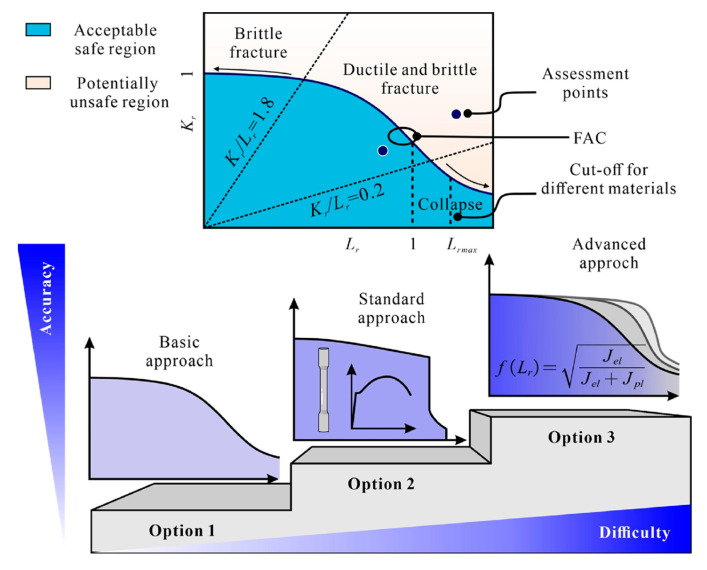
Schematic of Failure Assessment Diagram [19].

**Figure 7 materials-17-05677-f007:**
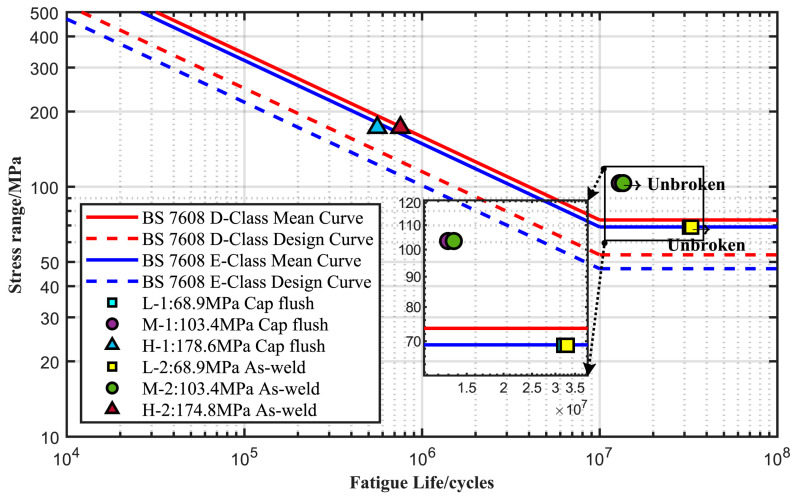
The plots of stress level versus fatigue life for six girth welds in relation to BS 7608: 2014 Class D and E curves.

**Figure 8 materials-17-05677-f008:**
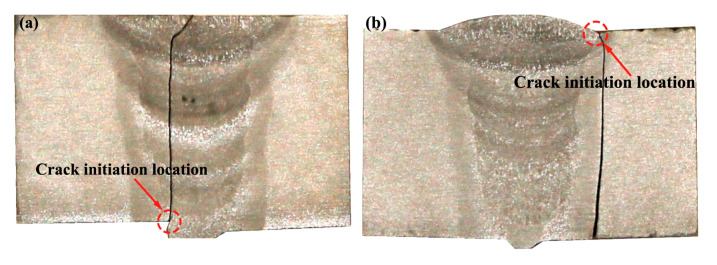
Crack growth direction along the wall thickness for girth welds with convex root: (**a**) H-1: cap flush; (**b**) H-2: as-weld.

**Figure 9 materials-17-05677-f009:**
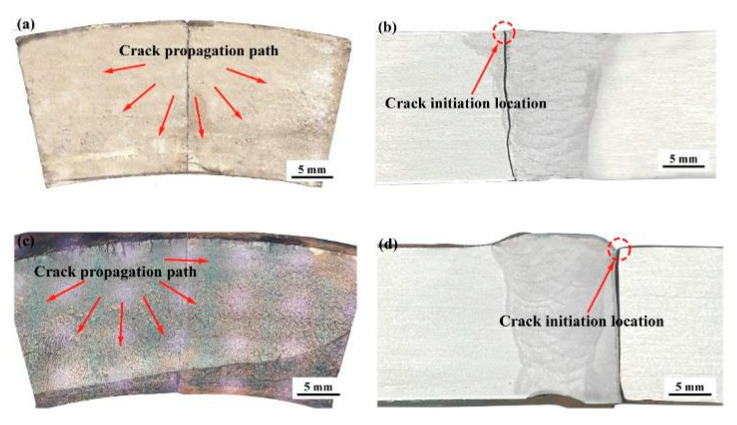
Crack growth path along the wall thickness and crack initiation location for girth welds with concave root: (**a**,**b**) H-3: cap flush; (**c**,**d**) H-4: as-weld.

**Figure 10 materials-17-05677-f010:**
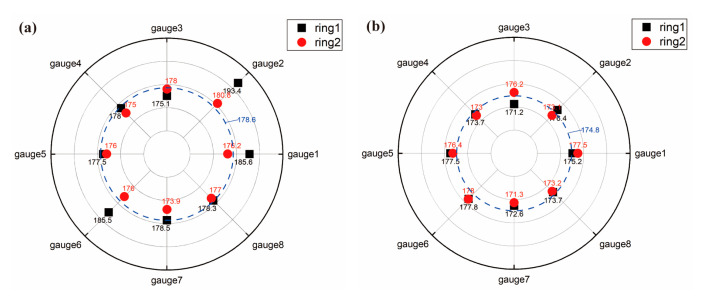
Average stress range distribution at both ends of the girth weld at the internal water pressure leak: (**a**) specimen H-1; (**b**) specimen H-2 (unit: MPa).

**Figure 11 materials-17-05677-f011:**
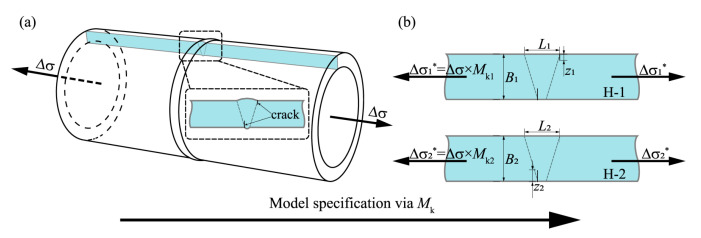
The model simplification approach: (**a**) Cross-section of circumferential weld in the actual pipeline; (**b**) Considering the stress concentration effect due to weld reinforcement by a correction factor, $M_{k}$.

**Figure 12 materials-17-05677-f012:**
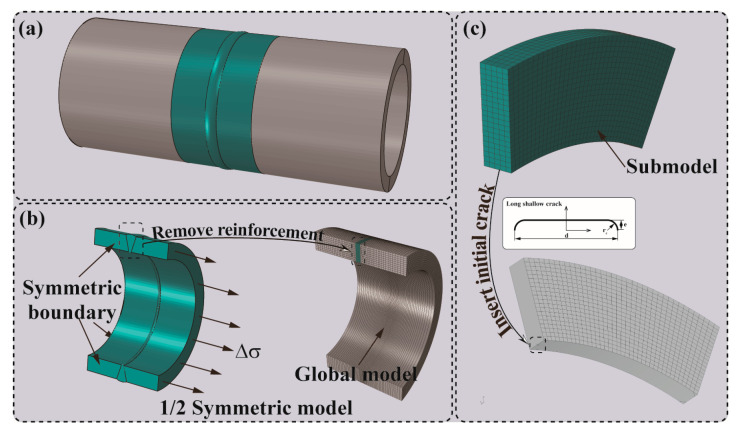
(**a**) An actual pipeline; (**b**) Details of the 1/2 symmetric model; (**c**) Insertion of initial crack.

**Figure 13 materials-17-05677-f013:**
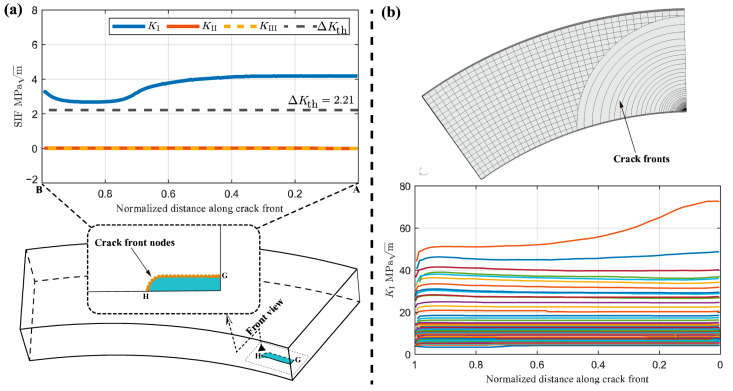
(**a**) The SIFs along initial crack front; (**b**) along crack front for the propagation of crack.

**Figure 14 materials-17-05677-f014:**
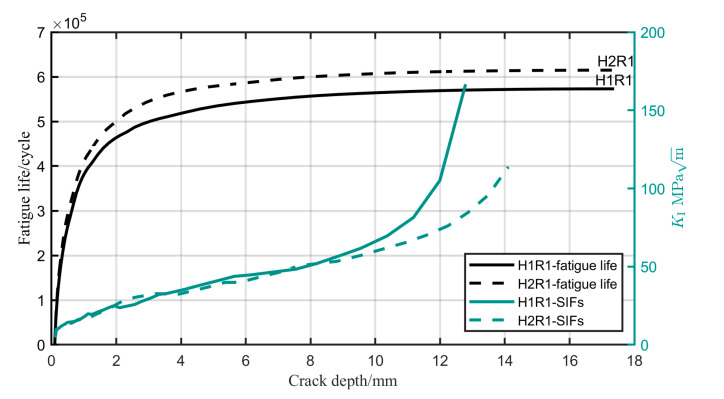
The variation trend of fatigue life and SIFs with the increasing crack depth.

**Figure 15 materials-17-05677-f015:**
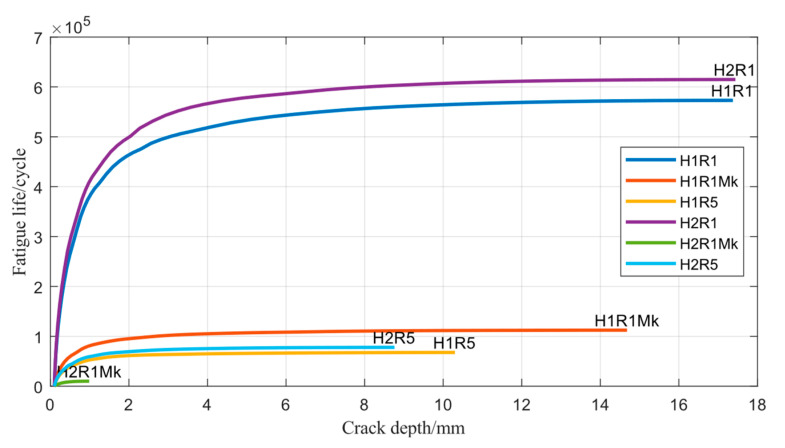
Number of cycles as a function of crack depth on the path for 50% normalized distance in the crack front (H1 and H2 represent specimens H-1 and H-2; R1 and R5 represent stress ratio of 0.1 and 0.5, respectively; Mk represents stress correction factor).

**Figure 16 materials-17-05677-f016:**
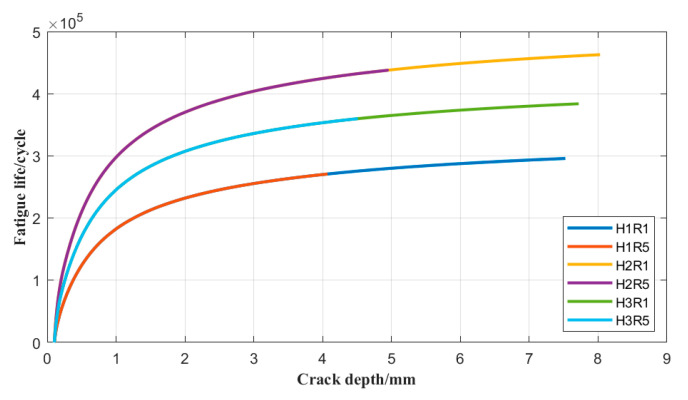
Fatigue life versus crack depth in the BS 7910 assessment considering misalignment.

**Figure 17 materials-17-05677-f017:**
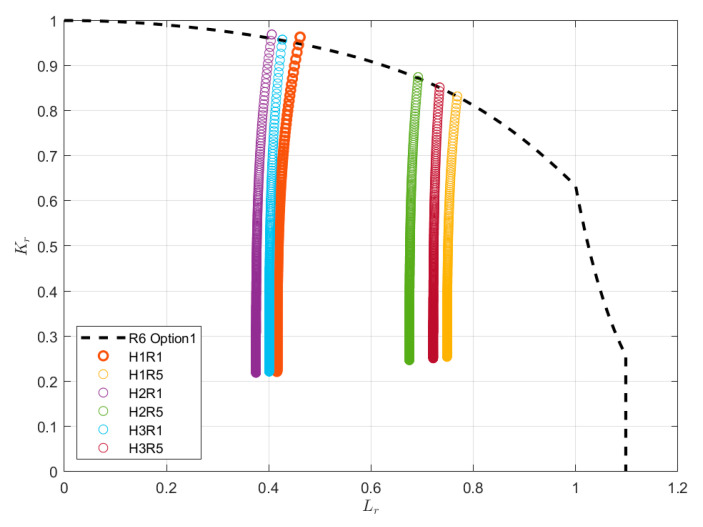
The iteration of fracture assessment points with welding misalignment.

**Table 1 materials-17-05677-t001:** The welding parameters of SCR girth weld.

	Current/A	Voltage/V	Feeding Speed /cm/min	Welding Speed /mm/min	Interlayer Temperature/°C
Double-sided Root welding	MIG:200 TIG:280	MIG:18 TIG:10	/	270	70
Fill	300	14	75	110–120	200–250
Weld cap1	230	13	45	120	220
Weld cap2	220	13	45	110	225

**Table 2 materials-17-05677-t002:** The parameters of X65 steel required in the NASGRO equation.

Parameters	Value	Parameters	Value
*σ*_u_/MPa	580	*a*_0_/mm	0.0381
*σ*_s_/MPa	506	*n*	2.8
*K*_Ic_/MPa·m^1/2^	109.9	*c*	1.23 × 10^−12^
Δ*K*_0_/MPa·m^1/2^	6.59	*P*	0.5
*A* _k_	0.75	*q*	0.5
*B* _k_	0.5	*C*th	2.2

**Table 3 materials-17-05677-t003:** The results of stress versus cycle life by resonance bending fatigue test.

Specimen No.	Stress Range Δσ/MPa	Internal Pressure σ_mean_/MPa	Loading Cycle	Post-Weld Treatment
L-1	68.9	72	32,003,250 (run-out)	Cap flush
L-2	68.9	65	32,900,204 (run-out)	As-weld
M-1	103.4	68	12,965,535 (run-out)	Cap flush
M-2	103.4	71	13,555,804 (run-out)	As-weld
H-1	172.4	67	559,600	Cap flush
H-2	173.0	67	755,400	As-weld
H-3	193.0	70	1,150,000	As-weld
H-4	193.0	70	3,050,000	Cap flush

**Table 4 materials-17-05677-t004:** The calculation results of $M_{k}$ at the axial loading mode.

Specimen	Crack Location	*L*/mm	*B*/mm	*z*/mm	*L*/*B*	*z*/*B*	*v*	*w*	*M* _k_
H-1	Weld root toe	3.2	18.3	0.1	≤2	≤0.05(L/B)^0.55^	0.3185	−0.31	1.6
H-2	Weld cap toe	14.1	18.3	0.1	≤2	≤0.05(L/B)^0.55^	0.4753	−0.31	2.4
H-3	Weld cap toe	/	/	0.1	/	/	/	/	1.0
H-4	Weld cap toe	15.5	18.3	0.1	≤2	≤0.05(L/B)^0.55^	0.4876	−0.31	2.5

**Table 5 materials-17-05677-t005:** FEA model specifications.

Specimen	Crack Location	*M* _k_
H-1	Weld root	1.0
H-1	Weld root	1.6
H-1	Weld root	1.0
H-2	Weld toe	1.0
H-2	Weld toe	2.4
H-2	Weld toe	1.0
H-3	Weld toe	1.0
H-4	Weld toe	2.5

**Table 6 materials-17-05677-t006:** Predicted fatigue life at R = 0.1 and 0.5 by numerical method.

Specimen	Stress Ratio	*M* _k_	Fatigue Life/Cycle
H-1	0.1	1	573,012
	0.5	1	67,642
	0.1	1.6	112,353
H-2	0.1	1	614,744
	0.5	1	77,834
	0.1	2.4	10,011
H-3	0.1	1	551,000
	0.5	1	69,000
H-4	0.1	1	556,452
	0.5	1	71,872
	0.1	2.5	9144

**Table 7 materials-17-05677-t007:** Misalignment parameters.

Specimen	Fracture Position	Welding Misalignment/mm
H-1	Inner	1.008
H-2	Outer	0.272
H-3	Outer	0.0
H-4	Outer	0.0

**Table 8 materials-17-05677-t008:** The results of the final failure size of crack and fatigue life predicted by BS 7910.

Specimen No.		Stress Ratio/*R*	Failure Crack Size/mm		Fatigue Life *N*/Cycle
			*a*	2*c*	
H-1	H1R1	0.1	7.31	17.74	295,000
	H1R5	0.5	3.99	9.25	270,000
H-2	H2R1	0.1	7.85	18.52	462,000
	H2R5	0.5	4.86	11.14	437,000
H-3	H3R1	0.1	7.52	17.41	382,000
	H3R5	0.5	4.42	10.03	358,000

**Table 9 materials-17-05677-t009:** Percentage deviation from experiment results across different assessment methods.

Specimen No.	FEA			FAD
	Stress Ratio/*R*	*M* _k_	Percentage Deviation/%	Percentage Deviation/%
H-1	0.1	1.0	+2.38	−47.29
		1.6	−79.93	
H-2	0.1	1.0	−18.63	−38.85
		2.4	−98.67	
H-3	0.1	1.0	−52.09	−66.78
H-4	0.1	1.0	−81.76	−66.78
	0.1	2.5	−99.70	

## Data Availability

The original contributions presented in this study are included in the article. Further inquiries can be directed to the corresponding author.

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
