# Peer review of "Steel Catenary Riser Fatigue Assessment: Fracture Mechanics Approach Versus S–N Curve Method"

_materials, 2024, doi:10.3390/ma17225677_

Round 1
Reviewer 1 Report
Comments and Suggestions for Authors
The manuscript presents the test results of the full-scale SCR girth weld. The manuscript is well organised. The references are properly chosen. The results of the calculations support the conclusions. However, the authors mentioned in the title and in the text that fatigue life was calculated using the S-N curve, but it does not occur in the text. Estimating fatigue life from the S-N curve from Figure 7 was not explained. It looks like fracture mechanics was used. The simplest way is using the Basquin equation. Also, there are shortcomings as the following:
1. Line 39 is repeated in line 40. It must be corrected.
2. Chapter 2.1. What is the accuracy of the test equipment? What range of force can be applied?
3. Fig. 10. The diagrams are too low resolutions.
4. Line 271. It is the statement that “ the difference in the stress range at different strain gauge positions is negligible”. However, in Figure 10 a) the difference for gauge 2 is 7%. In my opinion, it is not negligible.
5. Line 353. A reference missing.
6. Chapter 4.3.2. Why it was used another software? The ABAQUS and FRANC3D are not enough?
7. Chapter 5 (d). The variable Mk is not italic.
Author Response
Thank you very much for taking the time to review this manuscript. We greatly appreciate the valuable and constructive suggestions, which have significantly improved the quality of our work. I have provided detailed, point-by-point responses to each of your comments. Please see the attachment for further details.

Reviewer 2 Report
Comments and Suggestions for Authors
“Steel catenary risers fatigue assessment: fracture mechanics approach versus S-N curve method”,
manuscript materials-3291808,
by Zhang Niantao et al.
The manuscript is carefully written and contains interesting examples of using high cycle fatigue S-N approaches, as well as fatigue crack growth approaches, to model fatigue life of a structure. In my view the main issue of the paper is the need for further explanations on the connection of the real structure and the tests / analysis performed by the present authors. The problem consists on modelling possible failure of a submerged pipe exposed to loading under sea, (see Figure 1). Indeed Figure 1 shows a very complex problem, where wind effects, waves and even undersea ground stiffness may play a role; in those circunstances, it would be expected to find in the manuscript some description of the adopted strategy to deal with life prediction. Is it a probabilistic loading? cycles counting in that case? are there effects of cyclic plastic deformation? in that case how would that influence the results?
As editorial matters I indicate the following, in order to help the authors to improve their manuscript.
Line 39 – one sentence is repeated ( ‘ .... It is essential to assess fitness-for-service of SCR structures to ascertain if a known .... ‘ )
Line 42 – ‘ ... designed service life .... ‘ – if possible, it would be useful to readers to know details of what is the expected life duration in cycles, or years, or both, for this type of structure.
Line 82 - ...... 109.89 MPa sqrt m1/2 – this suggest a precision that fracture toughness measurements never have. It would be better to use just one decimal.
Line 97 – .... 5530 long – should be...... length (to keep ..... long, then please renove .... of)
Line 121 – this is a rather trivial expression for axial normal stress. Are there effects of internal pressure ? Lamé’s equations are required ? if so how are they accounted for in the analysis ?
Line 161 – it is .... and, not ..... And (this is not the start of a new paragraph, thereofr no capital should be used).
Section 3.1, starting in line 176 – somewhere it could be mentioned that the proposed FAD technique has its origin in the work of the British CEGB and its R6 assessment procedure.
Line 190 – please give a reference in support of the claim that BS 7910 is the most widely used engineering code to evaluate ..... etc. ....
Line 252 – ‘ .... for modelling models ....’ – please re-write, this is inadequate presentation.
Line 324 – is this Sih’s MTS (maximum tangential stress) criterion – please check the acronym !
Line 441 – not a ..... pint, but a ..... point.
Line 346 – not ......founding but ...... finding.
Line 402 – not ......failure, but .....to have failed.
Line 508 – please do not use capitals only in the title of journal articles (for consistency of presentation).
Finally, I suggest that in the title page tha names of the authors are given in the usual format – i.e., given name first, followed by family name (if you do the reverse, you will not be correctly indexed in bibliomeric databases)
Author Response

(The authors gave the same response as above.)

Reviewer 3 Report
Comments and Suggestions for Authors
The paper is interesting, having experimental, analytical and numerical parts.
There are some elements, more of form that need to be corrected. These have been highlighted in yellow in the PDF file and I have made the necessary comments for corrections.
If these corrections are made responsibly, I will accept the paper in the following form.

Author Response

(The authors gave the same response as above.)

Round 2
Reviewer 1 Report
Comments and Suggestions for Authors
The answers were satisfied for numbered comments. However, the authors have not answered the comment in the first paragraph: ‘However, the authors mentioned in the title and in the text that fatigue life was calculated using the S-N curve, but it does not occur in the text. Estimating fatigue life from the S-N curve from Figure 7 was not explained. It looks like fracture mechanics was used.”
The manuscript cannot be accepted, because the title does not match the content in the text.
Author Response
I sincerely apologize for my oversight in not addressing your question in the first paragraph. Thank you for giving me the opportunity to correct this mistake, and I greatly appreciate your understanding and patience. Your valuable and constructive suggestions have significantly contributed to enhancing this work, and I am grateful for the insights you have provided. Please see the attachment for further details.

Round 3
Reviewer 1 Report
Comments and Suggestions for Authors
The authors corrected the manuscript according to the recommendation. I accept to publish.